# A High-Throughput Small-Angle X-ray Scattering Assay to Determine the Conformational Change of Plasminogen

**DOI:** 10.3390/ijms241814258

**Published:** 2023-09-19

**Authors:** Adam J. Quek, Nathan P. Cowieson, Tom T. Caradoc-Davies, Paul J. Conroy, James C. Whisstock, Ruby H. P. Law

**Affiliations:** 1Department of Biochemistry and Molecular Biology, School of Biomedical Sciences, Monash University, Clayton, VIC 3800, Australia; 2Diamond Light Source Ltd., Diamond House, Harwell Science and Innovation Campus, Didcot OX11 0DE, UK; 3Australian Synchrotron, ANSTO_Melbourne, 800 Blackburn Rd., Clayton, VIC 3168, Australia

**Keywords:** SAXS, conformational change, plasminogen, fibrinolysis, structure-function, lysine binding site, lysine analogue, kringle domain

## Abstract

Plasminogen (Plg) is the inactive form of plasmin (Plm) that exists in two major glycoforms, referred to as glycoforms I and II (GI and GII). In the circulation, Plg assumes an activation-resistant “closed” conformation via interdomain interactions and is mediated by the lysine binding site (LBS) on the kringle (KR) domains. These inter-domain interactions can be readily disrupted when Plg binds to lysine/arginine residues on protein targets or free L-lysine and analogues. This causes Plg to convert into an “open” form, which is crucial for activation by host activators. In this study, we investigated how various ligands affect the kinetics of Plg conformational change using small-angle X-ray scattering (SAXS). We began by examining the open and closed conformations of Plg using size-exclusion chromatography (SEC) coupled with SAXS. Next, we developed a high-throughput (HTP) 96-well SAXS assay to study the conformational change of Plg. This method enables us to determine the *K*_open_ value, which is used to directly compare the effect of different ligands on Plg conformation. Based on our analysis using Plg GII, we have found that the *K*_open_ of ε-aminocaproic acid (EACA) is approximately three times greater than that of tranexamic acid (TXA), which is widely recognized as a highly effective ligand. We demonstrated further that Plg undergoes a conformational change when it binds to the C-terminal peptides of the inhibitor α2-antiplasmin (α2AP) and receptor Plg–R_KT_. Our findings suggest that in addition to the C-terminal lysine, internal lysine(s) are also necessary for the formation of open Plg. Finally, we compared the conformational changes of Plg GI and GII directly and found that the closed form of GI, which has an N-linked glycosylation, is less stable. To summarize, we have successfully determined the response of Plg to various ligand/receptor peptides by directly measuring the kinetics of its conformational changes.

## 1. Introduction

Fibrinolysis is essential for hemostasis and vascular patency. Plasmin (Plm) is the critical enzyme that breaks down fibrin clots. This enzyme also plays other vital roles in many physiological and pathological processes, including extracellular matrix degradation, tissue remodeling, wound healing, pathogen invasion, and cancer migration [1,2].

Plasminogen (Plg) is the zymogen form of Plm. It is made up of 791 residues and seven domains, including an N-terminal Pan apple domain (PAp), five kringle domains (KR1-5), and a serine protease domain (SP). The Plg found in human plasma has two main glycoforms: GI and GII, of 92,000 and 89,000 Da, respectively. Both glycoforms are O-linked glycosylated at Thr_346_ and Ser_248_; GI has an additional N-linked glycan at Asn_289_ [3,4,5] located in KR-3. The two glycoforms vary in half-life, substrate specificity, and target affinity and play different functional roles [6,7,8].

When circulating in the body, the full-length Plg forms a compact “closed” conformation to prevent unintended activation and non-specific binding (Appendix A). We have shown using X-ray crystallography that the closed form is attained through multiple inter-domain interactions mediated by the lysine binding site (LBS) on the KR domains [9], where the PAp domain is indispensable (Appendix A). However, lysine or analogues in solution or target receptors and substrates can interfere with these interactions and cause Plg to transition from a closed to an open conformation. When Plg assumes the relaxed and elongated “open” form, the activation loop, which is concealed [10] in the closed form, becomes exposed (Appendix A). Accordingly, Plg can be activated by the tissue-type (tPA) and urokinase-type (uPA) Plg activators through a proteolytic cleavage between residues Arg_561_ and Val_562_ [11].

The transition from closed to open Plg involves a significant change in the size and shape of the molecule. This characteristic makes it an ideal candidate for biological small-angle x-ray scattering (SAXS) studies, allowing detailed analysis of the transition process and its flexibility profiles in solution [12,13,14,15,16].

This paper describes a comprehensive study of the closed and open structures of Plg GII, a prominent glycoform found in human plasma.

In addition, we will detail the setup and validation of a high-throughput (HTP) SAXS assay that utilizes a 96-well plate, automated sampling, and a static mode for data collection, as previously described [17]. This method provides a reliable way to directly measure the kinetics of conformational transition in Plg. We used this method to compare the efficacy of different ligand-induced transformations in Plg GII conformation in solution, including L-lysine, lysine analogues EACA (ε-aminocaproic acid), and TXA (tranexamic acid).

Due to their significant scattering–interfering signals, SAXS is not a suitable technique for studying the conformational transitions of Plg when it binds to macromolecular protein molecules, such as fibrin, protease inhibitors, or cell surface receptors. In this study, we overcome this issue using peptides derived from two Plg binders: plasmin inhibitor α2-antiplasmin (α2AP) and Plg receptor Plg-R_KT_.

Several laboratories have utilized comparable techniques to study the transition of Plg from its closed to its open state. However, previous research was performed on preparations consisting of both GI and GII. Therefore, it remains to be investigated if there is any difference between conformational transition kinetics between GI and GII [10,18,19,20]. Accordingly, we performed a thorough comparative analysis of Plg GI and GII and found that GII is more stable than GI, as expected.

## 2. Results

### 2.1. Characterization of Closed-to-Open Plg GII

In our earlier studies using X-ray crystallography, we revealed how the LBS of KRs interacts with lysine or arginine residues [9] on the surface of the neighboring domains within the same molecule. These interactions lead to the formation of the closed Plg, as shown in Appendix A. However, the interactions between domains break up readily when LBSs form new interactions with external lysine and arginine residues on the surface of other molecules, such as receptors and fibrin, or L-lysine and analogues in solution. As a result, Plg transforms into an open form [9,18,21].

Figure 1 shows the result of an SEC-SAXS experiment (data collected immediately after size-exclusion chromatography) conducted on native human Plg GIIs. The exclusion volume profile of closed and open Plg GIIs (Figure 1a, as shown in the normalized *A*_280_) are readily distinguishable on an SEC column. To generate the fully open Plg GII, 10 mM EACA was used, as described in Materials and Methods. The two conformations have a comparable molecular mass (~90,000 Da), as reflected by the zero-angle scattering *I*(0). The radius of gyration (*R*_g_) calculated via Guinier approximation is 31.7 ± 0.11 Å for the closed and 48.8 ± 0.9 Å for open conformations (Figure 1b). Such a significant difference in *R*_g_ between the two conformations confirms that SAXS is indeed the method of choice for studying the kinetics of Plg confirmational transition.

The SAXS data reveal that the closed Plg is a well-defined globular structure. Accordingly, the closed Plg GII showed a relatively shallow gradient at the low angle with a clear transition (Guinier knee) to the high angle (Figure 1b). This is further confirmed by the bell-shaped curve on the dimensionless Kratky plot (Figure 1c). The pairwise distribution function *P*(r) is a skewed bell curve, from which the maximum diameter (*D*_max_) is determined to be 105 Å (Figure 1d). In the Porod plot (Figure 1e), the curve plateaus around 0.08 Å^−1^ roughly approximates a spheroidal particle with a diameter of 78 Å.

On the other hand, the open Plg GII form is a disordered and flexible structure. Accordingly, the data reveal a steep slope at the low angle (Figure 1b) without a well-defined transition between the lower and higher angles. There is also a lack of distinct maxima in the dimensionless Kratky plot **(Figure 1**c). Meanwhile, the *P*(r) function reveals a *D*_max_ of 170 Å (compared with 105 Å in the closed form (Figure 1d)) and without a plateau in the Porod plot (Figure 1e).

To further understand the population dynamics of Plg, we analyzed the SAXS data using the ensemble optimization method (EOM) [22,23] (Figure 1f) with the high-resolution crystal structure of individual domains of Plg [9]. Here, an in silico model was prepared in which flexible linkers of appropriate length joined each of the seven Plg domains (PAp, five KRs, and SP domains), assuming no additional interaction other than the interdomain linker between domains and a disulfide linkage between KR-2 and KR-3. A generic algorithm was used to select subpopulations from this random pool that, when taken together, best fit the SAXS data. As shown in Figure 1f, the *R*_g_ distribution of the random pool consists of a broader peak with a large maximum dimension of 60 Å. The selected ensemble for the closed form is compact and relatively homogeneous, as evidenced by the narrow peak in the distribution plot. In contrast, the open form has a broader peak that is shifted to the right of the random ensemble pool, indicating that in the open form, Plg GII is more elongated than would be expected from purely random domain movement (Appendix A), as illustrated in Appendix A, which shows a gallery of 12 possible open Plg models. This peak is significantly broader than the closed form, suggesting a more flexible structure. For a side-by-side comparison, ab initio low-resolution envelope models of the representative closed, and open Plg GII are shown (Appendix A). The unliganded Plg model resembles the crystal structure—compact and globular (see later discussions on the conformation stability of Plg GI and GII for further details)—whereas the EACA-bound Plg is elongated.

### 2.2. Conformational Transition Studies Using an HTP Assay

An HTP assay to study the kinetics of Plg conversion from a closed to an open state was established for this work using Plg GII (see Section 4 and Figure 2). Figure 3 shows the scattering curves and *R*_g_ generated from the Guinier analysis of Plg GII treated with EACA concentrations ranging from 0–20 mM.

The data showed that the transition from closed to open is readily observable (Figure 3a); all intermediate *R*_g_ values can be interpreted as a linear composite of the open and closed forms. The eigenvalues plot (Figure 3b,c) from singular value decomposition showed that the minimum number of components is two, namely fully open and fully closed, without any stable intermediates. To analyze the transition profiles of Plg recorded from the fully closed to the fully open forms, we fitted the kinetic data as follows: the *R*_g_ of Plg GII in 0 mM EACA (32 Å) was defined as closed. The *R*_g_ of Plg in 20 mM EACA (49 Å) was defined as open. The best fit for the kinetic data was achieved using a multi-site binding model, which yields a *K*_open_ of 0.35 mM. We use *K*_open_ to define ligand efficacy; it refers to the concentration of a ligand at which 50% of all Plg molecules in a solution transformed from the closed to the open conformation. If the closed conformation is more stable or the ligand is weaker, there will be a higher *K*_open_ value. The value of 0.35 mM found in this study is comparable to that of 0.45 mM previously reported on a mixed glycoform sample [6]. The slope of the Hill plot is 1.5 (Figure 3d inset), suggesting that cooperative binding of at least two EACA molecules is required for the conformational change of Plg.

### 2.3. Plg Conformational Transition in the Presence of L-Lysine and Peptides with a C-Terminal Lysine

We used the HTP assay detailed before to study the impact of free L-lysine residue alone and lysine-containing peptides derived from Plg-binding proteins, inhibitor α2AP, and cell receptor Plg-R_KT_**.**

Plg-R_KT_ is a 17 kDa transmembrane Plg receptor with a conserved lysine residue at the C-terminus [24,25]. Previous studies showed that Plg/Plm binds to a synthetic peptide called CK10, corresponding to a C-terminal peptide of Plg-R_KT_ [25].

α2AP is a specific and efficacious Plm inhibitor belonging to the SERPIN family. It inhibits free circulating Plm in the system and regulates the fibrinolysis [26]. In addition to the conserved SERPIN core, it has two unique extensions, one at the N- and one at the C-terminus. Plg/Plm binds to the C-terminal extension, which contains five conserved lysine/arginine residues, one of which is located at the extreme C-terminus. MK12 and NK55, corresponding to the last 12 and 55 residues (Figure 4), respectively, were used in this work [27].

We determine the *K*_open_ for Plg GII in the presence of L-lysine and these three peptides. All peptides have a C-terminal lysine, previously determined to be essential for Plg binding [24,28]. The results are shown in Figure 4; the *K*_open_ for L-lysine is 7.47 ± 1.12 mM, and CK10 is 0.73 ± 0.24 mM. Surprisingly, we could not detect any conformational change for MK12, even at the highest concentration tested at 10 mM.

Our data revealed that the concentration of free L-lysine required to induce a transition of Plg GII from closed to open is more than 20-fold that of EACA and ~10-fold that of CK10, confirming that L-lysine is a poor substrate. On the other hand, MK12 (_441_MEEDYPQFGSP***K***_452_) did not induce a conformational transition even at 10 mM, as mentioned previously, whereas CK10 (C_139_EQS***K***LFDS***K***_147_), which has only one internal lysine apart from the C-terminal lysine, is highly efficacious, with a *K*_open_ of 0.73 ± 0.24 mM. This data suggests that the internal lysine plays an indispensable role in CK10.

NK55 is also derived from the C-terminus of α2AP, but it has a total of seven internal lysine/arginine residues (_398_NPSAP***R***EL***K***EQQDSPGN***K***DFLQSL***K***GFP***R***GD***K***LFGPDL***K***LVPPMEEDYPQFGSP***K***_452_) [26]. The *K_open_* for NK55 is 0.08 ± 0.05 mM, which is significantly more efficacious than L-lysine (~100-fold) and CK10 (~10-fold). Although mapping the exact position of the internal lysine/arginine residues in CK10 and NK55 is beyond the scope of the current study, our findings strongly suggest that a minimum of one internal lysine residue is required for Plg GII to transition from its closed form to an open form. We also proposed that in the case of NK55, the internal lysine/arginine residues bind to the Plg KRs in the same way as a zipper, first leading to a conformational change followed by stabilizing and constraining the open form.

### 2.4. Conformation Stability of Plg GI and GII

We used SEC-SAXS to compare the conformation stability of Plg GI and GII **(**Figure 5), which were purified to homogeneity (Appendix A). Direct comparison of the scattering profile, Kratky plot, and *P*(r) functions of closed Plg GI and GII reveal that the *R*_g_ and *D*_max_ (in Å) for GI are 34.1 ± 0.4 and 109.5; and for GII are 31.8 ± 0.6 and 109.3, respectively. Based on these data, there is a subtle difference between closed GI and GII.

The ab initio low-resolution envelope models of GI and GII (average NSD values of 0.541 and 0.626, respectively) agree with the X-ray crystal structures (Figure 5f,g). CRYSOL analysis indicates that the experimental SAXS data are highly consistent with GI and GII crystal structures (χ values of 0.78 and 1.19, respectively). Interestingly, our previous X-ray crystallography studies on the GI and GII revealed that the two glycoforms are very similar except for the KR-3. In GI, KR-3 is N-linked glycosylated at Asn_289,_ and it is not visible in the electron density (Figure 5e–g), suggesting it is a flexible domain. In the current study, KR-3 is also poorly defined in the solution model. This observation supports our previous conclusion that the N_289_-glycosylation makes the KR-3 domain highly flexible. Based on our current observation, highly flexible domains in structures would not be accurately depicted in models generated from SAXS studies.

We proposed further that the relative flexibility of the KR-3 domain would significantly impact the overall conformational stability of Plg GI. Thus, we also performed kinetics studies on GI and GII to verify this hypothesis using EACA and TXA in the HTP format (Appendix A, Table 1).

Our data confirmed that the closed GII is more stable; the *K_open_* of EACA and TXA for GII is ~1.75 fold higher than for GI. Of these two ligands, EACA is weaker; the *K_open_* for both GI and GII is ~3 fold of TXA.

In comparison with L-lysine (Appendix A), based on the studies performed on Plg GII, the *K*_open_ for TXA (0.12 ± 0.008 mM) and EACA 0.35 ± 0.02 mM) are 60 and 20-fold lower (*K*_open_ for L-lysine is 7.47 ± 1.12 mM), respectively. These data, on the one hand, confirm the high efficacy of these anti-fibrinolytic therapeutics; they also permit a quantitative comparison of their performance. Importantly, our data support the notion that these lysine analogues can potentially outcompete the binding of Plg to its binding proteins Plg-R_KT_ and α2AP (Appendix A) at a therapeutic dosage in vivo (plasma concentration up to ~1 mM TXA and ~10 mM EACA).

## 3. Discussion

Plg is a vital therapeutic target for thrombotic and hemostatic diseases. Functionally, the closed Plg must undergo a significant structural change to become an open form upon binding to a target; this conformational change is also essential for its activation by the host activators to occur. The mechanism and process of activation and inhibition have been under intense scrutiny for many decades [6,19,21,28]. In this work, we used SEC-SAXS to fully characterize the closed and open conformations of native human Plg GII. This part of this work has provided crucial data to guide the setup of the HTP SAXS assays and validate the results obtained.

We used the HTP SAXS titration studies to show that the conformational change of Plg can be directly and readily measured. We also showed that *K_open_* correlates well, but reversely, with *K_activation_* by tPA (refer to Appendix A for direct comparison). Relevant to this field, the generation of Plm enzyme activity (*K_activation_*) is often used as a readout for ligand-induced conformational change. Compared to *K_open_,* the *K*_activation_ concentration is higher—>8-fold for EACA and >3.5-fold for TXA, respectively. Accordingly, we proposed that Plg activation by tPA may occur only after it is fully open.

Using the HTP SAXS assay, we have independently confirmed the previous observations that the transition from closed to open conformation is a single-step process [21] with multiple possible conformations, and this conformational change is triggered by positively cooperative binding to a minimum of two molecules of lysine or lysine analogues [10,21,28].

We have determined the *K*_open_ values for EACA, TXA, and L-lysine through ligand titration. This allows us to compare their ligand efficacies directly and accurately. Our findings indicate that TXA is the most efficacious ligand (Appendix A).

We also determined the *K*_open_ of small peptide ligands derived from the Plg-binding proteins Plg-R_KT_ receptor and inhibitor α2AP. Previous studies revealed that both Plg-R_KT_ and α2AP bind to Plg via the C-terminal lysine [24,29,30]; here, we showed that the additional internal lysine(s) is essential. Specifically, MK12-MEEDYPQFGSP***K*** from the last 12 residues of α2AP, which has a single C-terminal lysine residue, is not functional, whereas CK10-CEQS***K***LFSD***K*** from the last 10 residues of Plg-R_KT_, which has an additional internal lysine residue, is highly efficacious. This observation also aligns with our observation that the binding of at least two lysine residues is required for a conformational change (Figure 4). NK55, which consists of eight lysine/arginine residues, confers a higher efficiency in inducing conformational change, most likely via forming a more stable complex. Our findings indicate that having only a C-terminal lysine in a peptide is insufficient for Plg conformational change. The exact location of the necessary internal lysine will require further investigation.

Furthermore, Plasminogen-binding Group A Streptococcal M Protein (PAM) from Group A Streptococci has the highest reported binding affinity for Plg. In comparison, VEK35 (35 residues) and VEK75 (75 residues) peptides derived from PAM are extremely efficacious **(**Appendix A) [31]. Using the same HTP method we reported here for the first time, we showed that the dimeric VEK75 has a much higher activity than its shorter and monomeric counterpart, VEK35. Intriguingly, neither of these peptides has a C-terminal lysine. The binding of these peptides to Plg, and presumably PAM [31], would involve a different mode than that of Plg-R_KT_ and α2AP.

In this study, we also dissected, in full detail, the conformational stability of the two glycoforms. Our data revealed that the conformation of GII is more stable than that of GI. Previously, we hypothesized that the Asn_289_ N-glycan on GI enhances the mobility of KR3 and destabilizes its closed form [9]. Here, we generated the *K_open_* values for the direct comparison. We also showed that the superposed correlation between the X-ray crystal structures and the SAXS models is excellent; the disordered domain present in the crystal structure of GI (i.e., KR-3) is also poorly represented in the SAXS envelope. We proposed that in solution, the mobile domain provides weak diffraction and, therefore, becomes difficult to resolve from the background. This study enables a direct evaluation of how post-translational modifications affect the stability of the conformation in Plg.

In summary, we have developed and validated an HTP SAXS assay for Plg, which was used successfully to determine the kinetics and conformational changes of Plg in response to various ligand/receptor peptides.

## 4. Materials and Methods

### 4.1. Preparation of Plg Glycoform I and II

Plg was isolated and purified from human plasma from Red Cross Blood Bank Australia via three-step purification as previously described [3,9]. Plg GI and GII were prepared with a TXA gradient (Appendix A). Closed Plg was prepared via exhaustive dialysis of purified protein into the assay buffer (100 mM sodium phosphate pH 7.4, 5% glycerol). Proteins were concentrated to 1 mg/mL in centrifugal filters (Merck Millipore, Burlington, MA, USA). The final filtrate was kept as a solvent blank for SAXS experiments.

### 4.2. Preparation of Effector Ligands

EACA, TXA, and L-lysine were purchased from Sigma-Aldrich (St Louis, MO, USA), whereas peptides MK12, NK55, and CK10 were purchased from GL Biochem (Shanghai, China). Solutions of EACA, TXA, and L-lysine (0.5–1.0 M) were prepared in ultrapure water. The peptide ligands are slightly acidic and therefore were dissolved in 0.1 M NH_4_OH with gentle sonication. MK12 and CK10 were prepared as 20 mM stock solution, and NK55 was prepared as a 625 μM stock solution.

### 4.3. SAXS Data Collection

Experiments were performed at the Australian Synchrotron SAXS/WAXS beamline using a fixed energy of 12 keV and a camera length of 2.6 m.

For the SEC-SAXS experiment, in-line size exclusion chromatography using a WTC-010S5 column (WYATT) with a co-flow setup was used. For the HTP assays, 1 mg/mL protein samples were mixed 1:1 (*v*/*v*) with effector solutions to a final volume of 50 μL in a 96-well plate and incubated for 15–30 min before data collection at 16 °C. The final protein concentration for these experiments was 0.5 mg/mL in the assay buffer unless otherwise specified.

For each sample, a matching buffer was used for background scatter subtraction. To minimize the effect of radiation damage, 50 μL samples were flowed past the X-ray beam in a 1.5 mm diameter quartz capillary at 4 μL.s^−1^. Then, 15 to 20 2D scattering images were collected on a Pilatus 1M X-ray detector (Dectris, Baden-Daettwil, Swizerland) using 18 × 1 s exposures. The images were averaged, and the background was subtracted using the Scatterbrain software v.2.82 available at the Australian Synchrotron [32]

### 4.4. SAXS Data Analysis

Averaged SAXS data were processed with the ATSAS 2.7.2 software package (EMBL Hamburg, Germany). PRIMUS [33] was used for determining the radius of gyration (*R*_g_) and protein molecular weight via Guinier approximation [34] and the particle distance distribution function *P*(r) using indirect Fourier transform.

Ab initio models were generated with DAMMIN [35]. Ten models were generated and averaged for each dataset using DAMAVER [36]. The averaged model selected has a normalized spatial discrepancy (NSD) value equal to or less than 0.6. Atomistic models of open Plg were built using BUNCH. Five replicates were compared using DAMAVER with NSDs equal to or less than 1.65, and the most representative structure was used. The ensemble optimization method (EOM) was used to describe the activated Plg in terms of a population of related structures. SUPCOMB was used to superimpose the low-resolution ab initio models onto the x-ray crystal structures of Plg [37,38].

Singular value decomposition of the titration series data was carried out using the SVDPLOT program in the ATSAS suite of software [39].

### 4.5. SAXS Kinetic Analysis

For each titration series, the *R*_g_ values were normalized as follows:$$\mathrm{Normalized}\;Rg=\frac{Rg-\;\mathrm{minimum}\;Rg}{\mathrm{maximum}\;Rg-\;\mathrm{minimum}\;Rg}$$

Normalized *R*_g_ values were plotted against ligand or peptide concentration ([L]) and analyzed with the following specific binding equation using GraphPad Prism 6 (GraphPad, San Diego, CA, USA):$$\mathrm{Normalized}\;Rg=\frac{{\left[L\right]}^{h}}{{K_{open}}^{h}+{\left[L\right]}^{h}}$$
where h is the Hill slope and the kinetic constant, *K*_open_, is the ligand concentration at which 50% of Plg is in the open conformation (i.e., normalized *R*_g_ is 0.5).

### 4.6. Plg Activation Assay

The generation of Plm was monitored in 96-well microtiter plates using chromogenic substrate S-2251 (Chromogenix, Milan, Italy) after 1 h incubation of Plg with different concentrations of effector and tPA at 28 °C. Each reaction mix contained 0.25 μM Plg, 0.01 μM tPA, and the indicated amount of effector. After incubation, 0.2 mM of substrate S-2251 was added, and hydrolysis was monitored by continuous absorbance measurement at 405 nm (*A*_405 nm_). Initial rates of reaction (*V_0_*) were obtained by performing a linear regression of the first five minutes of the progress curve and were subsequently normalized as:$$\mathrm{Normalized}\;V_{0}=\frac{V_{0}-\;\mathrm{minimum}\;V_{0}}{\mathrm{maximum}\;V_{0}-\;\mathrm{minimum}\;V_{0}}$$
and plotted against the corresponding concentration of effectors ([L]). The resulting sigmoidal curves were fitted using:$$Normalized\;V_{0}=\frac{{\left[L\right]}^{h}}{{K_{activation}}^{h}+{\left[L\right]}^{h}}$$
where *h* is the Hill slope, and the kinetic constant *K*_activation_ is the ligand concentration at which the *V_0_* is 50% of the maximum.

## Figures and Tables

**Figure 1 ijms-24-14258-f001:**
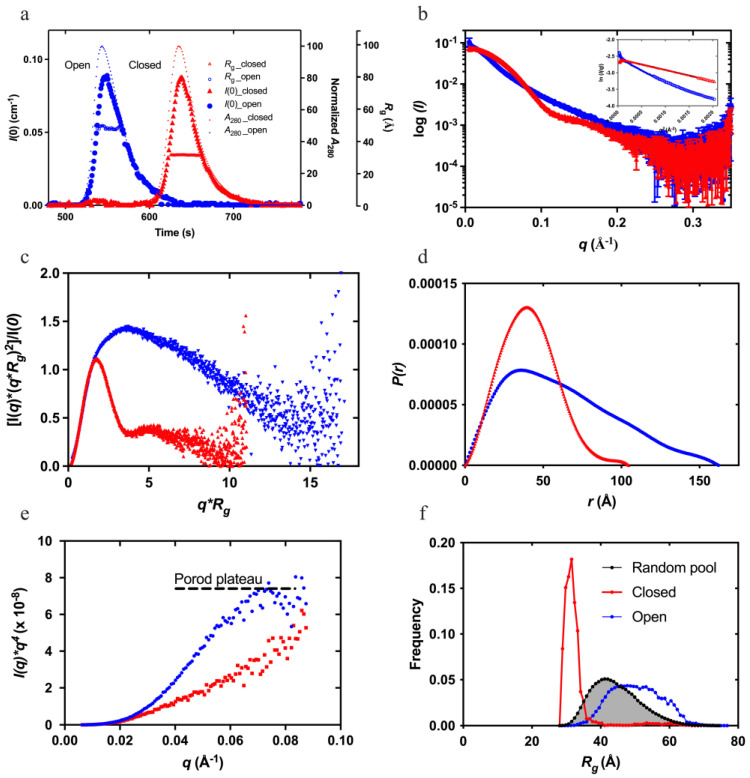
Small-angle X-ray scattering (SAXS) studies of plasminogen glycoform II (Plg GII)—showing the superposition of the closed (red) and open (blue) Plg GII: (**a**) coupled size-exclusion chromatography with SAXS (SEC-SAXS) profiles showing the *I*(0) (left *Y*-axis), normalized absorbance (A_280_), and *R*_g_ (right Y-axes); (**b**) SAXS profiles recorded; (**c**) dimensionless Kratky plot showing the globular nature of the closed and the disordered nature of the open conformation; (**d**) *P(r)* analysis shows the distinctive difference in the dimensions of the two forms; (**e**) Porod–Debye plots showing the Porod plateau of the closed form; and (**f**) ensemble optimization method (EOM) of closed and open Plg GII for analyzing the difference in *R*_g_ distribution. Distribution curves correspond to a random pool of 10,000 generated structures (grey) and the EOM-optimized ensemble of closed (blue) and open (red) Plg. The closed conformation is best represented by the compact distribution curve, and therefore, it is a homogenous ensemble. The distribution curve for the open form has shifted to the right; with a broader curve, it represents a more heterogeneous ensemble.

**Figure 2 ijms-24-14258-f002:**
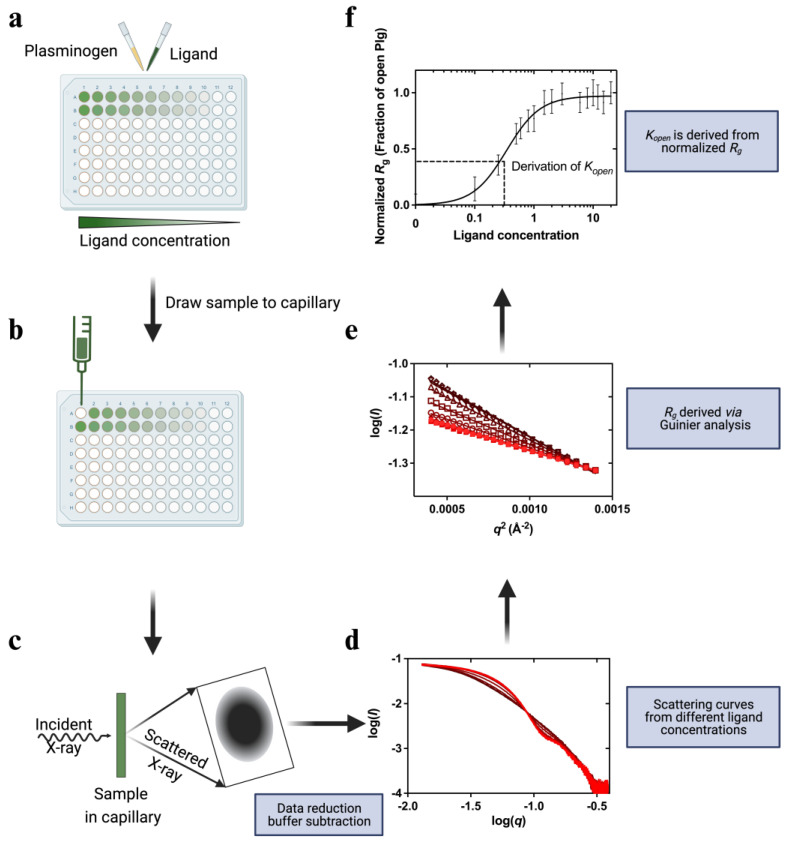
Overview of the high-throughput (HTP) kinetic studies of Plg conformational change. In this HTP assay, ligand-induced Plg conformational change is set up in a 96-well plate format. (**a**) Ligands at the study concentration are mixed with Plg and incubated for 30 min. (**b**,**c**) The samples flowed past the X-ray beam in a quartz capillary, and scattering images were collected. (**d**) The resulting 2D scattering profile is averaged and buffer subtracted. (**e**). *R_g_* values are derived via Guinier analysis (**f**) and then normalized before plotting against ligand concentration. The plot is fitted to a multi-site binding model, and *K_open_*, the ligand concentration required to induce 50% open Plg, is derived. *K_open_* is a kinetic parameter indicative of ligand efficacy or stability of closed conformation.

**Figure 3 ijms-24-14258-f003:**
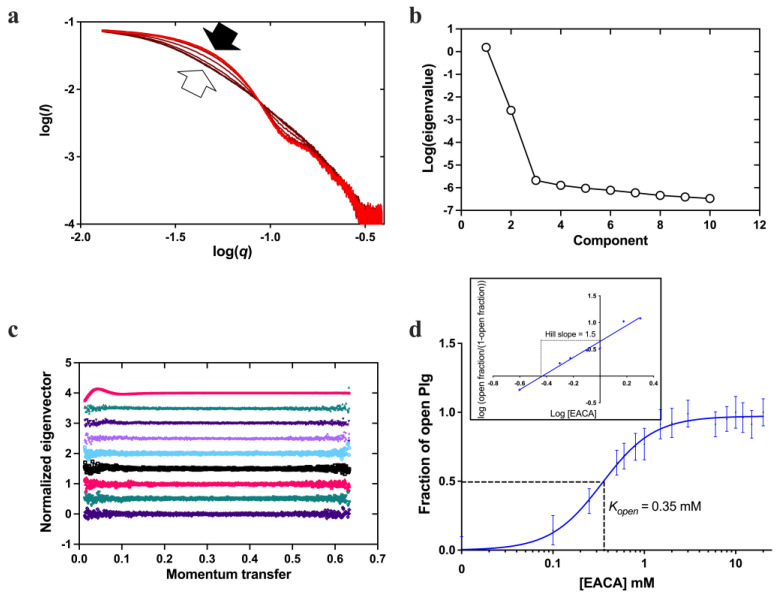
Conformational change of Plg GII in the presence of EACA. (**a**) X-ray scattering curves from the EACA titration. The curves are represented in a red color gradient, from closed (light red, solid arrow) to open form (dark red, blank arrow). (**b**) A plot of eigenvalues from singular value decomposition (SVD) analysis of scattering curves in the EACA titration series. The number of significant eigenvalues in the plot indicates the species contributing to the scattering data. (**c**) A plot of successive normalized eigenvectors from SVD analysis of scattering curves (colored) in (**a**). The first eigenvector is displayed at the top and the last at the bottom. (**d**) Fractions of Plg in the open form calculated from the SAXS scattering curves are plotted against EACA concentration. A multi-site cooperative kinetic curve is fitted and shown as a solid blue line. Inset: Hill plot showing positive cooperativity mechanism (hill slope = 1.5) of EACA binding.

**Figure 4 ijms-24-14258-f004:**
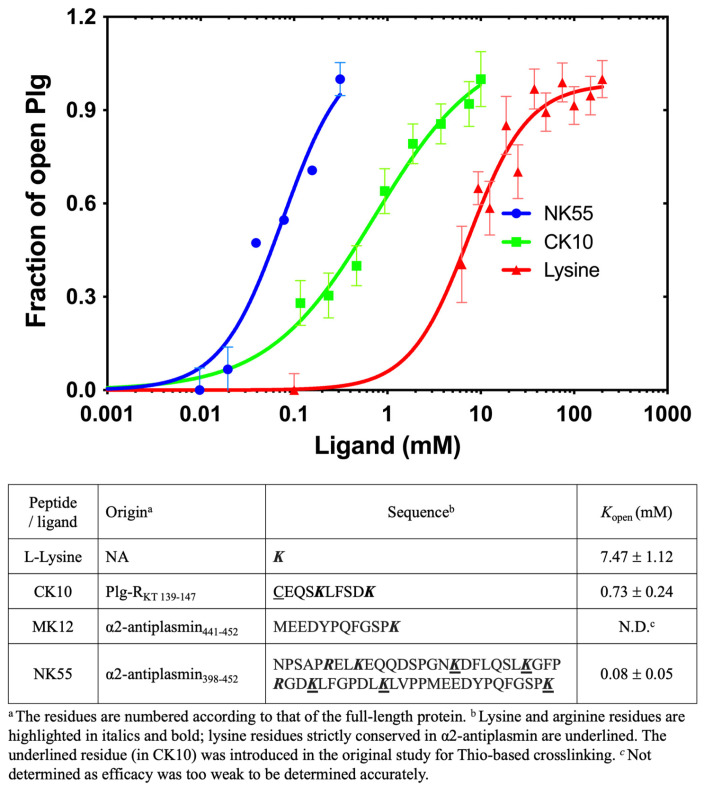
Conformational change of Plg GII in the presence of L-lysine and synthetic peptides derived from Plg-binding proteins. The graph shows the X-ray scattering titration curves of L-lysine (red) and synthetic peptides with a C-terminal lysine, namely CK10 (green) and NK55 (blue). MK12 was also tested, but at 10 mM, the highest concentration used, we did not observe any change in the SAXS profile of Plg GII. The *K*_open_ derived from this study and information on the peptides regarding the corresponding residue number in the proteins and the residue sequence are shown below. Lysine and arginine residues are highlighted in bold font.

**Figure 5 ijms-24-14258-f005:**
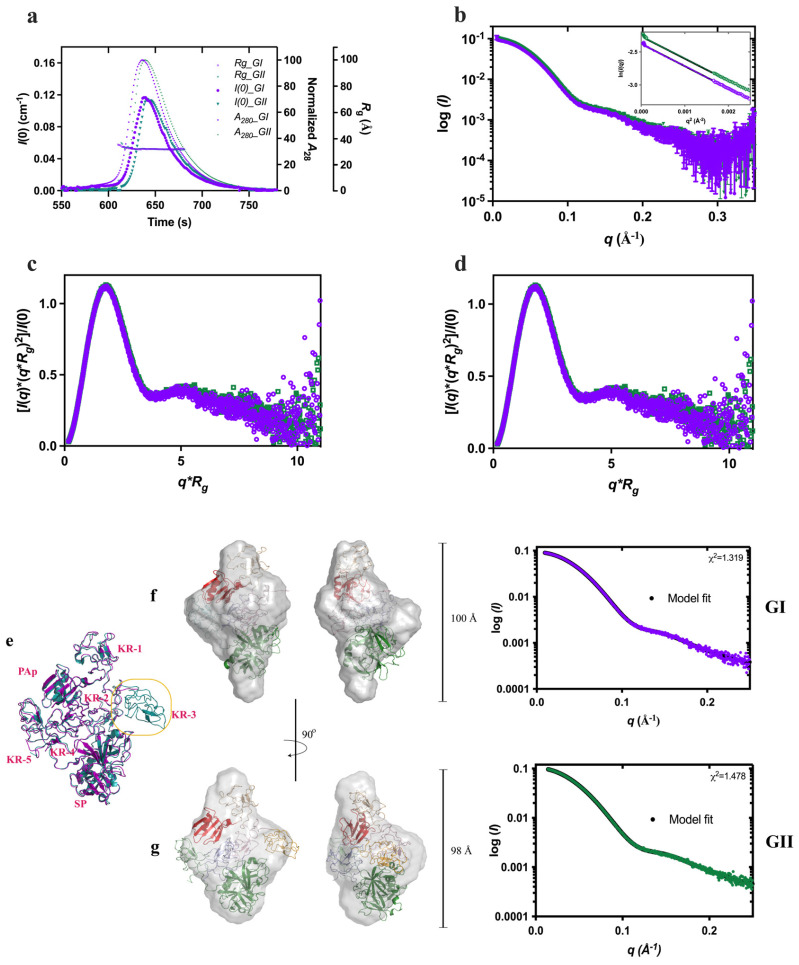
Comparison of Plg GI and GII conformations. Superposed of GI (purple) and GII (green): (**a**) SEC-SAXS profiles showing the *I*(0) (left *Y*-axis), normalized absorbance (A_280_), and *R*_g_ (right Y-axes); (**b**) SAXS profiles recorded; (**c**) dimensionless Kratky plot showing the globular nature and (**d**) *P(r)* analysis showing the similar dimensions of the two forms. (**e**) Superposition of Plg GI and GII crystal structures (PDB ID 4DUU and 4DUR, respectively, in the same color scheme as above). (**f**) Ab initio models of closed Plg GI and (**g**) GII; also shown are the correlation of scattering curves predicted from X-ray crystal structures using CRYSOL with the experimental data and the *χ^2^* values.

**Table 1 ijms-24-14258-t001:** *K*_open_ of plasminogen glycoform I and II (Plg GI and GII) in ε-aminocaproic acid (EACA) and tranexamic acid (TXA). *K*_open_ is the concentration of ligand required to induce the open conformation in 50% of the total Plg in solution (see Appendix A for more info).

Ligand	*K_open_* (mM)		
	Plg GI	Plg GII	Fold (GII/GI)
EACA	0.20 ± 0.01	0.35 ± 0.02	1.75
TXA	0.068 ± 0.002	0.120 ± 0.008	1.76

## Data Availability

Data is available to readers by directly emailing the corresponding authors.

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
