# Peer review of "A High-Throughput Small-Angle X-ray Scattering Assay to Determine the Conformational Change of Plasminogen"

_ijms, 2023, doi:10.3390/ijms241814258_

Round 1

Author Response

Point 1: In the in-silico modeling for the open structures of Plg, the authors assumed that there would be no interactions among the domains of Plg. Nevertheless, the explanation for the assumption regarding the absence of the inter-domain interactions within the modeling procedure remains insufficient. Therefore, a more comprehensive discussion about the underlying reasons and justifications for assuming the absence of the interactions during the modeling would enhance the readability of this paper.

Response 1: As per the reviewer's feedback, we agree that the assumption of no additional interdomain interaction in the open form may not be accurate as there is a lack of experimental supporting information. We also acknowledge that the open form could be a collective term for multiple open conformations. To address this concern, we have made the following changes:

  1. New Figure s1d shows 12 Representative models of open Plg from EOM analysis.
  2. P3 it reads: Here, an in-silico model was prepared in which flexible linkers of appropriate length joined each of the seven domains of plasminogen (PAp, five KRs and SP domains), assuming no additional interaction other than the interdomain linker between domains and a disulfide linkage between KR-2 and KR-3.
  3. P4: In contrast, the open form has a broader peak that is shifted to the right of the random ensemble pool, indicating that in the open form, Plg GII is more elongated than would be expected from purely random domain movement ( s1b-c); as illustrated in Figure s1d shows a gallery of 12 possible open Plg models.
  4. P6: Using the HTP SAXS assay, we have independently confirmed the previous observations that the transition from closed to open conformation is a single-step process (21) with multiple possible conformations

Point 2: Using the structural analysis based on the EOM, the authors selected the structural ensembles that best fit the scattering signals of the close and open forms. Comparing the Rg distributions obtained from these selected ensembles, they could identify significant structural differences between the close and open forms. However, they do not directly compare the conformational differences between the two forms in terms of their three-dimensional structure. To address this, it is suggested that the authors extract representative structures from the ensembles of both forms.

Response 2: We have included representative models of closed and open Plg in the new Figures s1b and c in response to the reviewer's comment. We also added in P7 of the MS to describe the method used to generate the figures:

Atomistic models of open Plg were built using BUNCH. Five replicates were compared using DAMAVER with NSDs equal to or less than 1.65 and the most representative structure was used. The ensemble optimization method (EOM) was used to describe the activated plasminogen in terms of a population of related structures. SUPCOMB was used to superimpose the low-resolution ab initio models onto the x-ray crystal structures of plasminogen (37,38).

Point 3: Furthermore, a more comprehensive discussion is needed regarding the differences in three-dimensional structure by comparing the representative structures. This approach would enhance the clarity of the study for readers.

Response 3: In response to the reviewer's comment, we have added the following text to P4 of the MS:

For a side-by-side comparison, ab initio low-resolution envelope models of the representative closed and open Plg GII are shown (Fig. s1b-c). The unliganded Plg model resembles the crystal structure – compact and globular (see later discussions on the Conformation stability of Plg GI and GII for further details); whereas the EACA-bound Plg is elongated.

Point 4: There is a missing period in the 65th line of this manuscript.

Response 4: The missing period is now added:

This paper comprehensively analyses the closed and open structures of Plg GII, a prominent glycoform found in human plasma.

Reviewer 2 Report

The paper by Quek et al. describes the conformational changes of plasminogen upon small ligand binding. SAXS is the main method of the paper. While it could be accompanied by computer simulations to gein deeper insights (see for instance SAXS publications of the Hummer lab), it is not necasssary. The results are already convincing using the old envelope method and Rg analysis. In addition, it is very well-written in clear and nice English. I only have very few comments (below) and I am happy to recommend this paper for publication.

Major comments

1) HT-SAXS (9+6-well plates) is actually not so big achievment, it was introduced at the Lawrence Berkeley National Laboratory at the SIBYLS Beamline like 15 years ago. This should be mentioned.

Minor comments:

1) Tranexamic acid

why capital T?

2) Figure 2, the font size should be increased, it is hard to read (especially compared to Figure 1)

3) same for Fig. 3B,C, subpanel of Figure 3D

4) Figure 4 uses different font that other figures (bold), the size and appearance of text in figures should be unified and larger

Author Response

Point 1: While it could be accompanied by computer simulations to gein deeper insights (see for instance SAXS publications of the Hummer lab), it is not necasssary. The results are already convincing using the old envelope method and Rg analysis.

Response 1: We appreciate your comments, and the critique is noted.

Point 2: HT-SAXS (9+6-well plates) is actually not so big achievment, it was introduced at the Lawrence Berkeley National Laboratory at the SIBYLS Beamline like 15 years ago. This should be mentioned.

Response 2: To address the Reviewer’s comment, we have:

  1. edited the Abstract, and now it reads: “Next, we developed a high throughput (HTP) 96-well SAXS assay setup to study the conformational change of Plg.”
  1. We have also incorporated a reference (#17, Hura et. al.) in the Introduction, and now it reads: “In addition, we will detail the setup and validation of a high-throughput (HTP) SAXS assay that utilizes a 96-well plate, automated sampling and static mode for data collection as previously described (17)”.

Point 3: Minor (1)Tranexamic acid: why capital T?

Response 3: WE have addressed the Reviewer’s comment by changing Tranexamic to tranexamic acid throughout the revised manuscript.

Point 4: Minor (2) Figure 2, the font size should be increased, it is hard to read (especially compared to Figure 1); Minor (3) same for Fig. 3B,C, subpanel of Figure 3D

Response 4: To address the Reviewer’s concern, we have now remade Figures 2 and 3 with a bigger font size and matching configurations.

Point 5: Figure 4 uses different font that other figures (bold), the size and appearance of text in figures should be unified and larger

Response 5: We have changed the fonts and style for Figures 1-4 where possible (some figures, such as Fig 3d, are best to remain as it is).